# Lactotransferrin Downregulation Drives the Metastatic Progression in Clear Cell Renal Cell Carcinoma

**DOI:** 10.3390/cancers12040847

**Published:** 2020-03-31

**Authors:** I-Jen Chiu, Yung-Ho Hsu, Jeng-Shou Chang, Jou-Chun Yang, Hui-Wen Chiu, Yuan-Feng Lin

**Affiliations:** 1Graduate Institute of Clinical Medicine, College of Medicine, Taipei Medical University, 250 Wu-Hsing Street, Taipei 11031, Taiwan; stirbar2000@yahoo.com.tw (I.-J.C.); tsubasa131047@gmail.com (J.-C.Y.); 2Division of Nephrology, Department of Internal Medicine, Shuang Ho Hospital, Taipei Medical University, New Taipei City 235, Taiwan; yhhsu@tmu.edu.tw; 3Division of Nephrology, Department of Internal Medicine, School of Medicine, College of Medicine, Taipei Medical University, Taipei 11031, Taiwan; 4Cancer Genome Research Center, Chang Gung Memorial Hospital, Linkou, Taoyuan 333, Taiwan; westlife828@gmail.com; 5Cell Physiology and Molecular Image Research Center, Wan Fang Hospital, Taipei Medical University, Taipei 11696, Taiwan

**Keywords:** clear cell renal cell carcinoma, lactotransferrin, LRP1, migration, metastasis

## Abstract

Clear cell renal cell carcinoma (ccRCC) is the main type of RCC, which is the most common type of malignant kidney tumor in adults. A subpopulation (>30%) of ccRCC patients develop metastasis; however, the molecular mechanism remains largely unknown. Here, we found that LTF, the gene encoding lactotransferrin, is dramatically downregulated in primary tumors compared to normal tissues derived from ccRCC patients deposited in The Cancer Genome Atlas (TCGA) database and is a favorable prognostic marker. Moreover, LTF downregulation appears to be more dominant in metastatic ccRCC. LTF overexpression suppresses migration ability in A498 ccRCC cells with high metastatic potential, whereas LTF knockdown fosters cellular migration in poorly metastatic ccRCC cells. Gene set enrichment analysis demonstrated that LTF expression inversely correlates with the progression of epithelial-mesenchymal transition (EMT) in ccRCC, which was further confirmed by RT-PCR experiments. Therapeutically, the administration of recombinant LTF protein significantly suppresses the cell migration ability and lung metastatic potential of ACHN cells, as well as LTF-silenced A498 cells. The gene knockdown of lipoprotein receptor-related protein 1 (LRP1) robustly blocked recombinant LTF protein-induced inhibition of cellular migration and gene expression of EMT markers in ACHN cells. LTF downregulation and LRP1 upregulation combined predicted a poor overall survival rate in ccRCC patients compared to that with either factor alone. Our findings uncover a new mechanism by which LTF may interact with LRP1 to inhibit metastatic progression in ccRCC and also reveal the therapeutic value of recombinant LTF protein in treating metastatic ccRCC.

## 1. Introduction

Renal cell carcinoma (RCC) accounts for ~90% of all renal tumors, and its incidence has been steadily increasing by 2–4% each year [1]. The major subtype of RCC is clear cell RCC (ccRCC), which accounts for approximately 75% of RCCs [2]. Over 30% of RCC patients exhibit metastasis at diagnosis [3]. High-grade RCC has a markedly high incidence of adrenal, nodal, hepatic, and pulmonary metastases [4]. Patients with metastatic RCC have a median survival of ~13 months and a 5-year survival rate <10% [5]. Until now, the efficacy of chemotherapy in patients with advanced RCC has not yet been satisfactory [6]. Sunitinib, which is an oral multitarget tyrosine kinase inhibitor, has been used to treat metastatic RCC [7]. Sunitinib interferes with and inhibits angiogenesis by inhibiting the vascular endothelial growth factor receptor and ligand (VEGFR/VEGF) [8]. Sunitinib can prolong overall survival in metastatic RCC, and approximately 70% of patients show an initial response [9,10]. However, RCC patients treated with sunitinib eventually become resistant [10]. More recently, immunotherapy (alone or in combination with TKIs) and cabozantinib exhibited a relative higher objective response rate and have been recommended as the standard of care for first-line treatment in metastatic RCC [11]. Therefore, there is an urgent need to more effectively treat metastatic RCC.

Lactotransferrin (LTF), also known as lactoferrin, plays an important role in innate immune defense [12]. LTF, which has iron-binding ability, was first discovered in mammary secretions but is synthesized by most mammalian tissues [13]. It is mainly found in breast milk and is also present in a variety of secretions derived from epithelial cells and the secondary granules of neutrophils [14]. Previous studies indicated that LTF is involved in many important biological functions and possesses antibacterial, anticancer, antiviral, anti-inflammatory, antifungal, and immune regulatory activities [15,16,17]. Some reports have demonstrated that the iron-binding ability of LTF and the interaction between LTF and its specific receptors are responsible for various physiological activities [16,18,19]. LTF interacts with receptors on target cells, including LDL receptor-related protein 1 (LRP1/CD91), C-X-C-motif cytokine receptor 4 (CXCR4), omentin-1 (intelectin-1), and Toll-like receptor 4 (TLR4) [20]. Evidence indicates that LRP1 expression is observed mostly in highly aggressive prostate cancer [21]. *LTF* expression in breast cancer correlates with the life expectancy of patients and important clinical and physiologic features of the disease [22]. In cancer therapy, LTF attenuates cell growth and invasion in several cancers [17,23,24]. Furthermore, LTF inhibits osteosarcoma cell proliferation and migration by regulating LRP1 and NF-kB p65 [25]. LTF can induce apoptosis and cause cell cycle arrest in breast cancer [26]. In addition, LTF inhibits epithelial-to-mesenchymal transition (EMT) and induces mesenchymal-to-epithelial transition (MET) in oral squamous cell carcinoma [24]. However, the effects of LTF in RCC are not clearly understood.

The aims of this study were to evaluate the role of the LTF gene in ccRCC and to investigate the possible mechanism. The results suggest that LTF may predict the outcome of ccRCC. LTF downregulation increases cellular migration ability and triggers the EMT progression of ccRCC. Moreover, LTF treatment effectively suppresses the metastatic potential of ccRCC cells by targeting LRP1. LTF merits further investigation as a potential diagnostic marker and therapeutic strategy for ccRCC patients.

## 2. Results

### 2.1. LTF Downregulation Is Commonly Found and Is Related to a Poor Prognosis in ccRCC

We examine the transcriptional profile of *LTF* in normal tissues and primary tumors derived from TCGA patients with clear cell, chromophobe and papillary RCC. The data showed that *LTF* mRNA levels in primary tumors were significantly (*p* = 1.2 × 10^−11^) lower than those of normal tissues in the TCGA ccRCC dataset (Figure 1A,B). This view was not predominant in TCGA chromophobe (Appendix A) and papillary (Appendix A) RCC datasets. In 72 paired normal and tumor tissues from RCC patients, the *LTF* mRNA levels in most of the paired samples were downregulated in primary tumors (Figure 1C). Accordingly, the protein levels of LTF in primary tumors were relatively lower than those in paired normal tissues derived from ccRCC patients (Figure 1D). Moreover, Kaplan–Meier analyses of TCGA RCC patient data under a maximal risk condition as described in Materials and Methods demonstrated that low *LTF* expression in primary tumors or disease classified as ccRCC was correlated with a poor overall survival rate (Figure 1E). Specifically, patients with ccRCC expressing a low level of LTF transcript had the shortest overall survival time (Figure 1E). We further found that TCGA ccRCC patients with primary tumors expressing a high level of LTF transcript had a 72.2% 5-year survival rate, while patients with primary tumors harboring a low level of LTF transcript had a 23.1% 5-year survival rate (Figure 2A). Kaplan–Meier analysis of recurrence-free survival probability showed that TCGA ccRCC patients with primary tumors expressing a high LTF transcript levels exhibited an 85.5% 5-year recurrence-free survival rate, while this rate decreased to 71.8% in ccRCC patients with primary tumors expressing a low level of LTF transcript (Figure 2B). In addition, the proportion of primary tumors expressing a low level of LTF transcript was extensively detected in TCGA ccRCC patients who were female or had higher pathologic stages (Figure 2C). Nevertheless, the proportion of primary tumors showing low and high LTF transcript levels stratified by age and pathological grade was not significantly different (Figure 2C). The transcriptional profiling of LTF in ccRCC with different pathologic stages revealed that LTF expression gradually declined from stage I to stage IV primary tumors (Figure 2D). Another Kaplan-Meier analysis of LTF protein levels determined by the intensities (scores 0 and 1 represent the low expression levels and scores 2 and 3 represent high expression levels) IHC staining indicated that a decreased level of LTF protein was also highly associated with a poor overall survival rate (Figure 2E,F).

### 2.2. LTF Downregulation Enhances Cellular Migration Ability and Triggers the Progression of Epithelial-Mesenchymal Transition in ccRCC

Since metastatic spread is frequently found in the late pathological stage, we next analyzed the mRNA levels of LTF in primary tumors derived from TCGA ccRCC patients diagnosed without or with cancer metastasis. The data showed that LTF mRNA is relatively lower in primary tumors from ccRCC patients with cancer metastasis (Figure 3A). On the other hand, the endogenous mRNA levels of LTF in ACHN ccRCC cells exhibiting a high migration ability were relatively lower than those in A498 ccRCC cells showing a poor migration ability (Figure 3B,C). While the overexpression of the exogenous LTF gene in ACHN cells dramatically suppressed cellular migration ability (Figure 3D–F), the gene knockdown of endogenous LTF using two independent shRNA clones predominantly enhanced the migration ability of A498 cells (Figure 3G–I).

To ascertain the molecular mechanism by which LTF downregulation promotes ccRCC metastasis, we utilized the Gene Set Enrichment Analysis (GSEA) program to simulate the possibly activated pathways. TCGA ccRCC patients with primary tumors exhibiting a low level of LTF transcript and lymph node metastasis or a primary tumor harboring a high level of LTF transcript but without lymph node metastasis were selected for Pearson’s correlation tests against the coexpression of *LTF* and other somatic genes. The ranking results from Pearson’s correlation tests were then input into the GSEA program to predict potentially activated or inhibited pathways under these two pathological conditions (Figure 4A). We found that nine signaling pathways were significantly (*p* < 0.001) predicted to be activated in metastatic ccRCC with low LTF expression but inhibited in nonmetastatic ccRCC with high LTF expression (Figure 4B). These pathways are involved in cellular EMT, proliferation (E2F targets, Myc targets and G2/M checkpoint), DNA repair, TNF-alpha-NF-κB signaling, apical junctions, and apoptosis (Figure 4B). Among these candidates, the transcriptional profile of the hallmark gene set, which reflects the activity of cellular EMT from the Molecular Signatures Database (MSigDB), appeared to be highly activated upon LTF downregulation in metastatic ccRCC, as demonstrated by the highest enrichment score, but predominantly inhibited in nonmetastatic ccRCC with high LTF expression, as demonstrated by the lowest enrichment score in the GSEA simulation (Figure 4C). In the primary tumors derived from metastatic ccRCC with LTF downregulation, CDH6 expression was positive, but ENO2 expression was negatively correlated with LTF expression (Figure 4D). PT-PCR and Western blotting experiments indicated that ACHN cells expressed a higher level of ENO2 transcript and protein but a lower level of CDH6 mRNA and protein than A498 cells (Figure 4E and Appendix A). While the enforced expression of the exogenous LTF gene in ACHN cells reduced ENO2 mRNA and protein expression but increased CDH6 mRNA and protein expression (Figure 4F and Appendix A), silencing endogenous LTF expression in A498 cells enhanced ENO2 mRNA and protein expression but reduced CDH6 mRNA and protein expression (Figure 4G and Appendix A). In addition, Kaplan–Meier analyses revealed that higher expression of the EMT gene set indicated a poor overall survival rate in TCGA ccRCC patients (Figure 4H). Moreover, the signature combining high expression of the EMT gene set with low LTF transcript level was correlated with a poor prognosis in ccRCC patients (Figure 4H). In addition, we also examined E2F DNA-binding activity by a luciferase-based reporter assay and cell cycle changes by propidium iodide-based flow cytometric analysis. The data showed that LTF knockdown in A498 cells enhanced E2F DNA-binding activity, which was suppressed by adding recombinant LTF (recLTF) protein into the cell culture (Appendix A). Nevertheless, there were no significant changes in the cell number of Sub-G1 phase, including apoptotic cells, in the PI-based flow cytometric analysis (Appendix A) even though GSEA simulation predicted an increased activity of apoptosis-related gene set upon LTF downregulation in ccRCC. 

### 2.3. LTF Administration Effectively Suppresses the Metastatic Potential of ccRCC Cells Probably by Targeting LRP1

We next estimated the therapeutic effectiveness of recLTF protein on highly metastatic ACHN cells in vitro and in vivo. Cell migration assays showed that the inclusion of recLTF protein dose-dependently suppressed the migration ability of ACHN cells (Figure 5A,B). Similar views were also found in another ccRCC cells, Caki2 (Appendix A). Moreover, the administration of recLTF protein significantly (*p* = 2.3 × 10^−6^) inhibited the lung metastatic potential of ACHN cells (Figure 5C,D). Whereas treatment with LTF-specific antibody dose-dependently restored the migration ability of LTF-overexpressing ACHN cells (Figure 5E), the addition of recLTF protein dramatically suppressed the migration ability of LTF-silenced A498 cells (Figure 5F). This finding highlighted LTF as an independent factor that modulates the metastatic potential of ccRCC cells. 

LRP1 has been identified as an LTF receptor [27]. To delineate whether recLTF protein suppresses ccRCC metastatic progression through binding with LRP1, we performed gene knockdown of LRP1 in ACHN cells (Figure 6A). LRP1 knockdown appeared to compromise the recLTF protein-mediated suppression of cellular migration ability in ACHN cells (Figure 6B,C). Moreover, the inclusion of recLTF protein failed to alter the gene expression of ENO2 and CDH6 in the LRP1-silenced ACHN cells (Figure 6D). In addition, the transcriptional profiling of LRP1 in the TCGA ccRCC database revealed that LRP1 expression in primary tumors was significantly (*p* < 0.001) higher than that in normal tissues (Figure 6E–G). Kaplan–Meier analyses demonstrated that LRP1 acts as a poor prognostic marker, and the signature of combining high LRP1 level with low LTF expression predicts a poor overall survival rate in ccRCC patients (Figure 6H). 

## 3. Discussion

When patients present with recurrent metastatic RCC and no surgical options are suitable, systemic treatment should be considered. Although several agents now exist for the treatment of metastatic RCC, they general do not produce durable complete responses, and thus, there is an urgent need to improve treatment for metastatic RCC [28]. EMT is a complex trans-differentiation process and is an important component in the metastasis and invasion of cancers. Over 30% of RCC patients are diagnosed with metastasis, and patients with metastatic RCC have a very low 5-year survival rate [4,6]. In this study, the results revealed that LTF mRNA levels in tumors of ccRCC are significantly lower than those in normal tissues (Figure 1A,B). The observations are quite similar to those of paired normal and tumor tissues from RCC patients (Figure 1C,D). Furthermore, LTF downregulation predicts a poor prognosis in ccRCC (Figure 1E). In addition, ccRCC patients with primary tumors expressing a higher LTF transcript level exhibited a higher 5-year survival rate and recurrence-free survival probability than the primary tumors showing a lower LTF transcript level (Figure 2A,B). The proportion of primary tumors expressing a low level of LTF was detected in ccRCC patients who were female or had higher pathologic stages (Figure 2C). Therefore, LTF downregulation predicts a poor prognosis. Previous research has shown that the correlation between the expression of LTF in breast tumors and the life expectancy of patients is important. The 5-year survival rate of breast cancer patients was higher in the LTF-high group. LTF expression correlated positively with moderate differentiation grade and negatively with high differentiation grade in luminal B or basal tumors [22]. Another study reported the potential for LTF in chemoprevention and that it may be a biologically relevant prognostic marker of prostate cancer. LTF downregulation in prostate tumor cells is remarkably associated with prostate-specific antigen recurrence after radical prostatectomy [29]. In addition, ACHN and A498 cells used in this study were derived from male and female RCC patients, respectively. In comparison with ACHN cells, A498 cells expressed a higher LTF and exhibited a poorer metastatic potential. Although LTF is mainly found in breast milk, it seems to be that LTF downregulation likely tends to promote cancer progression in female patients with ccRCC and breast cancer. Therefore, quantitative characteristics of LTF expression have prognostic applicability in disease progression.

LTF has many important biological functions, including anticancer effects. LTF attenuates cell growth, migration and invasion in several cancers [17,24,25]. EMT is an important process in malignant cancer cells that causes invasion into the surrounding tissues and metastasis via lymph vessels or blood [30]. Previously, the authors reported that LTF caused the considerable disappearance of mesenchymal features, reversed polygonal epithelial morphology and induced extracellular E-cadherin expression [24]. Thus, restraining EMT and inducing MET in metastatic RCC cells should improve anticancer strategies. In the present study, the LTF mRNA level was low in primary tumors from ccRCC patients with cancer metastasis (Figure 3A). Furthermore, the low endogenous mRNA levels of LTF in ccRCC cells were related to a high migration ability (Figure 3B,C). To understand the molecular mechanism by which LTF downregulation promotes ccRCC metastasis, we utilized GSEA to identify possible pathways. We found that nine signaling pathways were significantly predicted to be activated in metastatic ccRCC with low LTF expression but inhibited in nonmetastatic ccRCC with high LTF expression (Figure 4B). Among these candidates, CDH6 expression was positively correlated and ENO2 expression was negatively correlated with LTF expression in primary tumors derived from metastatic ccRCC with low LTF (Figure 4D). Cadherin-6 (CDH6) is a type II cadherin that contributes to inducing apoptosis via dephosphorylation of ERK [31]. Goeppert et al. indicated that CDH6 downregulation was observed and that CDH6 might be a putative tumor suppressor in cholangiocarcinomas [32]. On the other hand, CDH6 was also reported to be upregulated in other cancer types [33,34]. Enolase 2 (ENO2) is a neuron-specific enolase and is primarily expressed by mature neurons. ENO2 plays a key role in glycolysis, promoting the conversion of β-glycerophosphate into dihydroxyacetone phosphate [35,36]. Recent evidence from a multivariate analysis showed that high ENO2 expression was an independent prognostic factor for acute lymphoblastic leukemia patients [37]. Another recent study concluded that high ENO2 expression was associated with poor overall survival in ccRCC patients [38]. However, there are no reports describing the relationship between ENO2, CDH6 and LTF in RCC. The signature combining high expression of the EMT gene set with low LTF expression was correlated with a poor prognosis in ccRCC patients (Figure 4H). RecLTF protein significantly inhibited cell migration and the lung metastatic potential of ACHN cells (Figure 5A–D). Treatment with LTF restored the migration ability of LTF-overexpressing ACHN cells (Figure 5E). In addition to EMT, a recent review article demonstrated that the systemic inflammation may damage the immune system, thereby allowing tumoral invasion [11]. These findings highlighted that LTF may inhibit the metastatic potential of ccRCC cells through the suppression of inflammation [17] and the enhancement of cancer immune surveillance [11]. However, further studies are necessary to evaluate the possible mechanisms leading to the anti-metastatic properties of LTF in RCC cells.

LRP1 is an LTF receptor [27]. Upregulation of LRP1 has been reported in various human diseases, and LRP1 plays a critical role in the regulation of many cellular processes, including survival, cell proliferation, motility, and differentiation [39]. Previous studies have demonstrated that tissue-type plasminogen activator binds to LRP1 and activates extracellular signal-regulated kinases (Erk)1/2 to stimulate matrix metalloproteinase (MMP)-9 production and induce the EMT of kidney cells [38]. Additionally, LRP1 has been shown to be upregulated in various cancers and promotes cancer cell invasion, migration and tumor progression through the activation of focal adhesion kinase (FAK), Akt and Erk1/2 pathways [40,41,42]. In this study, silencing LRP1 inhibited the recLTF protein-suppressed cellular migration ability and failed to alter the gene expression of ENO2 and CDH6 in ccRCC cells (Figure 6B–D). In addition, LRP1 expression in primary tumors was significantly higher than that in normal tissues in the TCGA ccRCC database (Figure 6E–G). The signature combining high LRP1 level with low LTF expression is a poor prognostic marker in ccRCC patients (Figure 6H).

## 4. Materials and Methods

### 4.1. Clinical and Molecular Data for RCC Patients

The clinical data, including age, sex, cancer grade, cancer stage, TNM stage, and overall survival (OS) time for TCGA ccRCC patients, were collected from the UCSC Xena website [43]. The molecular data obtained by RNAseq (polyA þ Illumina HiSeq) analysis of the TCGA ccRCC cohort were also downloaded from the UCSC Xena website (Appendix A). The stratification of mRNA levels of LTF, CDH6, ENO2, and LRP1 into the low- and high-level groups was determined by using Cutoff Finder [44] under a maximal risk condition in Kaplan–Meier analyses because gene expression data are usually represented by metric or at least ordinal variables [45]. 

### 4.2. Cell Culture Condition

The human renal adenocarcinoma cell lines ACHN and A498 were purchased from American Type Culture Collection (ATCC); the cells were maintained in minimal essential medium (MEM) supplemented with 10% FBS, penicillin (100 U/mL) and streptomycin (100 μg/mL). All media and supplements were purchased from Gibco Life Technologies. 293T cells were cultured in DMEM with 10% FBS and incubated at 37 °C with 5% CO_2_. All cell lines were routinely authenticated on the basis of short tandem repeat (STR) analysis, morphologic and growth characteristics and mycoplasma detection.

### 4.3. Cellular Migration Assays

Cellular migration ability was assessed by using the Boyden Chamber Assay (NeuroProbe, Gaithersburg, MD, USA). Cells (1.5 × 10^4^) in serum-free culture medium without or with the designated concentrations of LTF antibody (Elabscience, Houston, TX, USA) or recombinant LTF protein (Sino Biological Inc., Wayne, PA, USA) were added to the upper chamber of the device on a membrane with an 8.0 μm pore and precoated with fibronectin (Invitrogen, Waltham, MA, USA), and the lower chamber was filled with 10% FBS culture medium. After a 2 (for ACHN cells) or 3 (for A498 cells)-hour incubation, the membrane was soaked in methanol for 10 min and stained with Giemsa, which was diluted 10-fold by double-distilled water, for 1 h. Then, the membrane was attached to slides, and the cells on the upper side of the membrane were carefully removed with a cotton swab. The cells on the lower side were photographed. The migrated cells were quantified by counting the cells in three random areas under a microscope at 400× magnification.

### 4.4. Lentivirus-Driven shRNA Infection 

All derivatives of the shRNA vector with a puromycin selection marker were obtained from the National RNAi Core Facility Platform in Taiwan. Lentiviruses were produced by cotransfecting the shRNA-expressing vector with the pMDG and p△8.91 constructs into 293T cells using a calcium phosphate transfection kit (Invitrogen). After incubation for 48–72 h, the media were collected as viral stocks. Cells (50% confluent) grown on 6-well plates were cultured in fresh media containing 5 μg/mL polybrene (SantaCruz, Dallas, TX, USA) before infection overnight with a lentiviral particle-driven control or candidate gene shRNA at 2–10 multiplicity of infection (MOI). To select cells stably expressing the control or candidate gene shRNA, cells were further cultivated in the presence of puromycin (10 μg/mL) for 24 h. Cell lysates from the puromycin-resistant cells were subsequently subjected to RT-PCR analysis to confirm the efficiency of gene knockdown.

### 4.5. Reverse Transcription PCR (RT-PCR) and Quantitative PCR (Q-PCR)

Total RNA was extracted from cells using a TRIzol extraction kit (Invitrogen). Aliquots (5 μg) of total RNA were treated with M-MLV reverse transcriptase (Invitrogen) and then amplified with Taq-polymerase (Protech) using paired primers (for LTF, forward-GACCTGTGGAAGGATATCTTGCTGTGGCGG and reverse-CACCGCCACAGCAAGATATCCTTCCACAGGTC; for CDK6, forward-GAAAACAGGGAGCAGTACCAAGTG and reverse-CATCAAACATATCCAGCCCCTC; for ENO2, forward-GTGGAGCAAGAGAAACTGGACAAC and reverse-CTTGAGTGTATGGTAGACCTCTGCAC; for GAPDH, forward-AGGTCGGAGTCAACGGATTTG and reverse-GTGATGGCATGGACTGTGGTC). Power SYBR^TM^ Green PCR Master Mix (Thermo Fisher) was used for Q-PCR. The mRNA levels were normalized to those of GAPDH. Fold changes were calculated using the 2^−ΔΔCt^ method.

### 4.6. Western Blotting Assay

Aliquots of 100 μg of total protein and HR Pre-Stained Protein Marker 10–170 kDa (Biotools, New Taipei City, Taiwan) were loaded into each well of an SDS gel, separated by electrophoresis and then transferred to PVDF membranes. The membranes were incubated with blocking buffer (5% skim milk in TBS containing 0.1% Tween-20) for 2 h at room temperature. The samples were incubated with primary antibodies against CDH6 (Genetex, Hsin-Chu, Taiwan #33279), ENO2 (Genetex, #124345), LRP1 (Signalway Antibody, College Park, MA, USA #48595) and GAPDH (AbFrontier, Seoul, Korea #PA0212) overnight at 4  °C. After extensive washing, the membranes were incubated with a peroxidase-labeled secondary antibody for 1 h at room temperature. Immunoreactive bands were visualized by using an enhanced chemiluminescence system (Amersham Biosciences, Tokyo, Japan). Uncut Blots can be found at Appendix A.

### 4.7. Immunohistochemistry Staining

Paraffin-embedded tumor sections (3 μm thickness) purchased from Super Biochips were heated and deparaffinized using xylene and rehydrated in a graded series of ethanol with a final wash in tap water. Antigen retrieval was accomplished with Target Retrieval Solution (DAKO, Woodbridge, VA, USA) in a decloaking chamber (Biocare Medical, Concord, CA, USA). Endogenous peroxidase activity was quenched by hydrogen peroxide. Sections were then incubated with anti-LTF antibody (Elabscience, Houston, TX, USA #14688; 1:100) at 4 °C overnight. A Vectastain ABC peroxidase system (Vector Laboratories, Burlingame, CA, USA) was used to detect the reaction products. 

### 4.8. Animal Experiments

NOD/SCID mice were obtained from the National Laboratory Animal Center in Taiwan and maintained in compliance with institutional policy. All animal procedures were approved (LAC-2018-0516) by the Institutional Animal Care and Use Committee at Taipei Medical University. For the in vivo lung metastatic colonization assay, 1 × 10^5^ cells in 100 μl PBS were implanted into the mice through tail vein injection. The mice were sacrificed, and the lungs were obtained for histological analysis. Metastatic lung nodules were quantified after staining with H&E using a dissecting microscope. 

### 4.9. Statistical Analysis

SPSS 17.0 software (Informer Technologies, Roseau, Dominica) was used to analyze statistical significance. One-way ANOVA and Tukey’s test were utilized to compare the differential display of *LTF* gene expression in normal tissues and primary tumors. Pearson’s correlation tests were performed to estimate the association between *LTF* levels and the detected parameters. Evaluation of the survival probabilities was determined by Kaplan–Meier analysis and the log-rank test. Nonparametric Mann–Whitney U test and Friedman test were used to analyze two independent samples and three or more related samples, respectively. In all analyses, *p* values of <0.05 were considered statistically significant.

## 5. Conclusions

In summary, LTF downregulation is commonly found in ccRCC and predicts a poor prognosis. LTF downregulation enhanced cellular migration ability and triggered the progression of EMT in ccRCC. LTF administration suppressed the metastatic potential of ccRCC cells, probably via LRP1. Furthermore, LTF expression positively correlated with CDH6 expression and negatively correlated with ENO2 expression in primary tumors derived from metastatic ccRCC with low LTF. LRP1 upregulation combined with decreased LTF expression predicts a poor overall survival rate in ccRCC patients. These findings imply that LTF may play an important role in the metastatic progression of ccRCC.

## Figures and Tables

**Figure 1 cancers-12-00847-f001:**
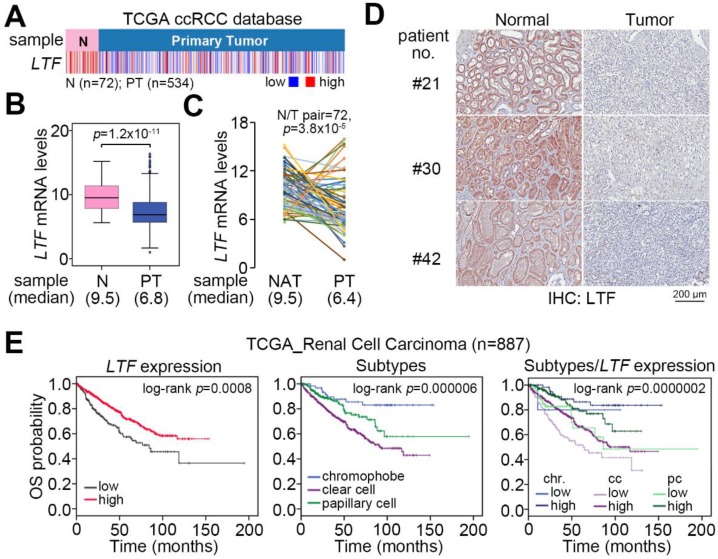
LTF downregulation is predominantly found in ccRCC and correlates with a poor prognosis. (**A**–**C**) Heatmap (**A**), boxplot (**B**) and points & connecting lines for the transcriptional profiling of *LTF* in normal (N) tissues/primary tumors (PT) and paired (*n* = 72) normal adjacent tissues (NAT)/PT, respectively, derived from TCGA ccRCC patients. (**D**) Representative immunohistochemistry (IHC) staining for LTF protein in paired normal/tumor tissues was derived from enrolled ccRCC patients. (**E**) Kaplan–Meier analyses of overall survival (OS) at a maximal risk condition for LTF mRNA levels [left; low (*n* = 201), high (*n* = 686)], RCC subtypes [middle; chromophobe (*n* = 65), clear cell (*n* = 533), papillary cell (*n* = 289)] and the combination of LTF mRNA levels and RCC subtype [right; chromophobe (chr): low (*n* = 5), high (*n* = 60); clear cell (cc): low (*n* = 102), high (*n* = 431); papillary cell (pc): low (*n* = 94), high (*n* = 195) in TCGA RCC patients.

**Figure 2 cancers-12-00847-f002:**
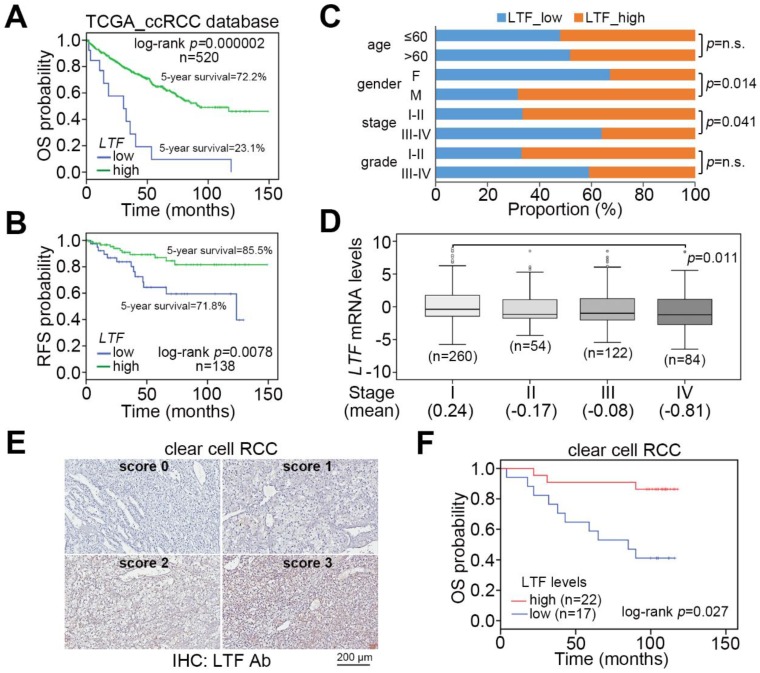
LTF downregulation correlates with cancer progression in ccRCC. (**A**,**B**) Kaplan–Meier analyses at a maximal risk condition of overall survival (OS, (**A**)) and recurrence-free survival (RFS, (**B**)) probability in TCGA ccRCC patients with low (*n* = 13 for OS, *n* = 39 for RFS) and high (*n* = 507 for OS, *n* = 99 for RFS) levels of *LTF* gene expression. (**C**) Histogram displays the results of a Chi-square test for the indicated *LTF* mRNA levels and pathologic variables including age, sex, stage, and grade obtained from the TCGA ccRCC database. The symbol “n.s.” denotes a nonsignificant result. (**D**) Boxplot represents the transcriptional profile of *LTF* gene expression in TCGA ccRCC samples with different pathological stages. The significant differences were analyzed by one-way ANOVA with Tukey’s test. (**E**) The IHC results for the LTF protein intensities ranged from a score of 0 to a score of 3. (**F**) Kaplan–Meier analysis of low (scores 0 and 1) and high (scores 2 and 3) LTF protein levels from the IHC analysis of tumors from 39 ccRCC patients.

**Figure 3 cancers-12-00847-f003:**
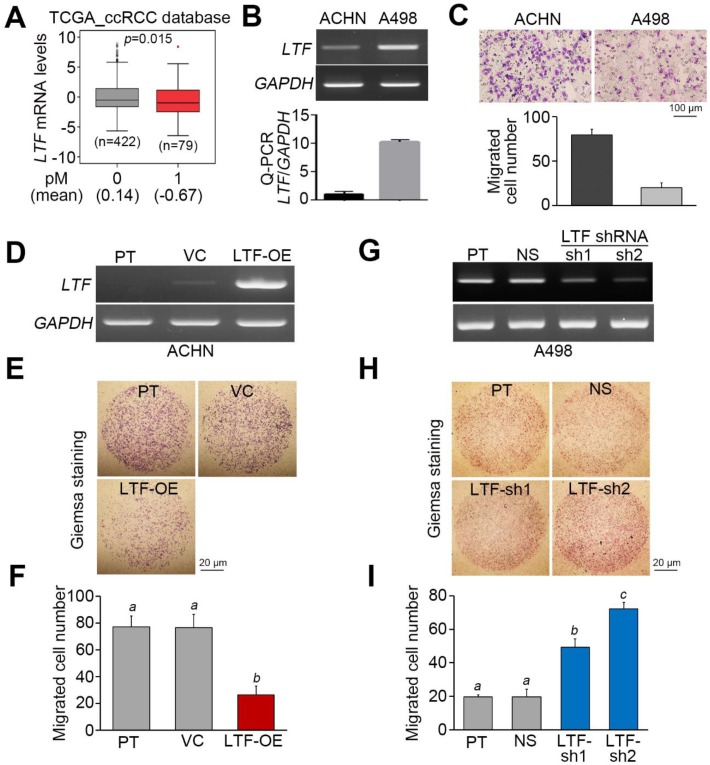
LTF expression negatively associates with cellular migration ability in ccRCC. (**A**) Boxplot for the transcriptional profile of LTF gene expression in primary tumors derived from TCGA ccRCC patients without (0) or with (1) pathologic M (pM) status. Student’s t-test was used to estimate the statistical significance. (**B**) LTF mRNA levels determined by RT-PCR (upper) and quantitative PCR (bottom) in the tested ccRCC cell lines, ACHN and A498. (**C**) Giemsa staining (top) and data from three independent experiments (bottom) analyzing migrated cells in the 3-h Transwell assay of ccRCC cells. (**D**–**F**) LTF mRNA levels determined by RT-PCR (**D**) and Giemsa staining and (**E**) a histogram representing data from three independent experiments (**F**) analyzing migrated cells after a 2-h Transwell assay in ACHN cells without (parental, PT) or with vector control (VC) or LTF overexpression (OE). (**G**–**I**) LTF mRNA levels determined by RT-PCR (**G**) and Giemsa staining and a (**H**) histogram representing data from three independent experiments (**I**) analyzing migrated cells after 4-h Transwell assay in A498 cells without (parental, PT) or with a stable transfection for nonsilencing (NS) control shRNA or two independent LTF shRNA (sh) clones. In (**B**,**D**,**G**), GAPDH was used as an internal control for RT-PCR experiments. The different letters in each column of (**F**,**I**) indicate statistical significance at *p* < 0.05 as determined by one-way ANOVA with Tukey’s test.

**Figure 4 cancers-12-00847-f004:**
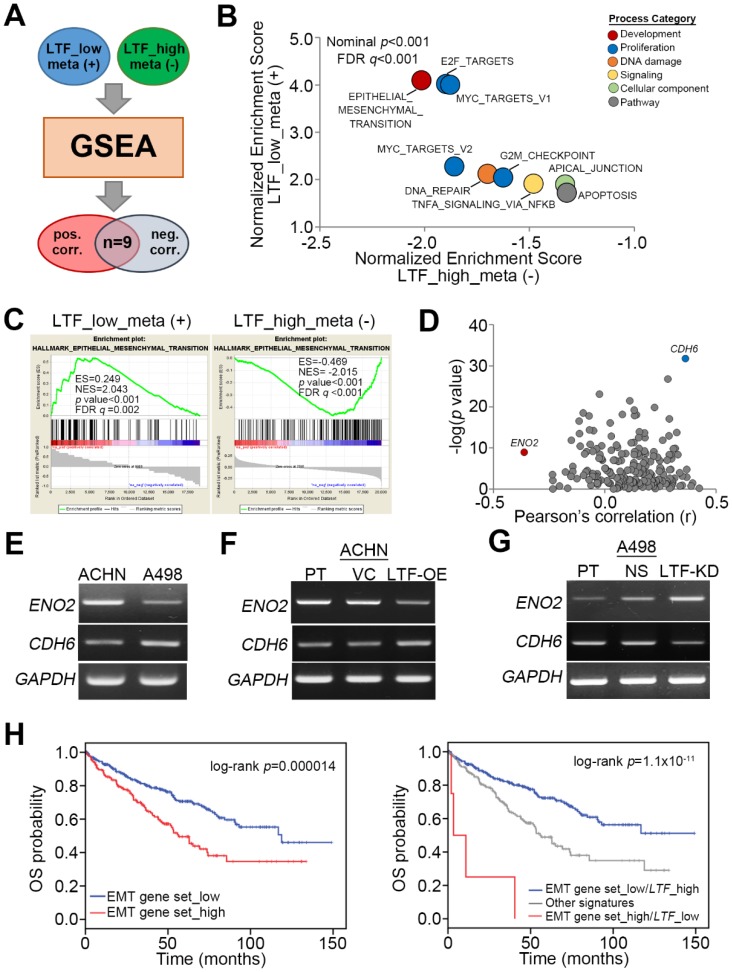
LTF expression inversely correlates with the progression of epithelial-mesenchymal transition in ccRCC. (**A**,**B**) Flowchart (**A**) and results (**B**) of GSEA analysis against Pearson’s correlation test for LTF co-expression with somatic genes in metastatic ccRCC with low LTF expression and nonmetastatic ccRCC with high LTF expression using TCGA ccRCC database. (**C**) The enrichment score (ES) derived from the correlation between the epithelial-mesenchymal transition gene set and the queried Pearson’s correlation coefficient (r) is plotted (green curve). NES and FDR denote the normalized enrichment score and false discovery rate, respectively. (**D**) Scatchard plots for Pearson’s correlation coefficient and the –log [*p* value] derived from Pearson’s correlation test for the mRNA levels between POLQ and the genes in the epithelial-mesenchymal transition gene set in metastatic ccRCC with low LTF expression. (**E**–**G**) RT-PCR analyses of ENO2, CDH6 and GAPDH gene expression in ACHN and A498 cells (**E**); ACHN cells without (PT) or with VC and LTF-OE (**F**); or A498 cells without (PT) or with NS and LTF-KD (**G**). GAPDH was used as an internal control for RT-PCR experiments. (**H**) Kaplan–Meier analyses of OS at a maximal risk condition for the mRNA levels of the epithelial-mesenchymal transition (EMT) gene set alone [left; low (*n* = 343), high (*n* =177)] or combined with LTF gene expression [right; EMT gene set_low/LTF_high (n=334), other signatures (*n* = 182), EMT gene set _high/LTF_low (*n* = 4)] in TCGA ccRCC patients.

**Figure 5 cancers-12-00847-f005:**
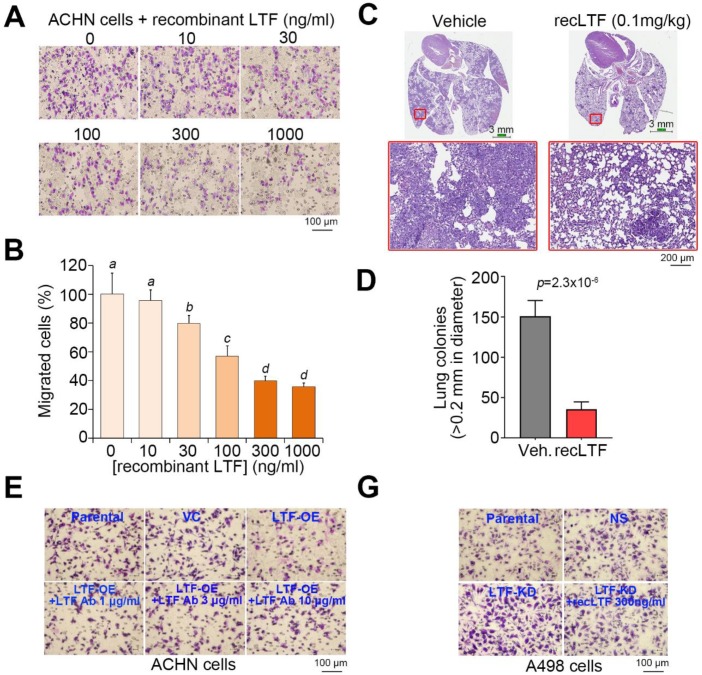
LTF inhibits the metastatic potential of ccRCC cells. (**A**,**B**) Giemsa staining (**A**) and histogram representing data from three independent experiments (**B**) analyzing migrated cells in the 3-h Transwell assay for ACHN cells pretreated with recombinant LTF (recLTF) protein at the indicated concentrations for 24 h. (**C**) Hematoxylin/eosin staining of lung tissues derived from mice post-transplantation with ACHN cells for 4 weeks without (vehicle, PBS) or with the administration of recLTF at 0.1 mg/kg twice a week. The red regions indicate the colonies of ACHN cells. Scale bars in the upper and lower pictures denote 3 and 0.2 mm, respectively. (**D**) Boxplot for the lung tumor colonies (>0.2 mm in diameter) counted from tumor-bearing mice (*n* = 5) as shown in C. Nonparametric Mann–Whitney U test was used to estimate statistical significance. (**E**,**F**) Giemsa staining (**E**) and histogram representing data from three independent experiments (**F**) analyzing migrated cells after a 2-h Transwell assay in parental and vector control ACHN cells and LTF-OE ACHN cells pretreated without or with LTF antibody (Ab) at the designated concentrations for 24 h. (**G**,**H**) Giemsa staining (**G**) and histogram representing data from three independent experiments (**H**) analyzing migrated cells after a 3-h Transwell assay in parental and vector control A498 cells and LTF-silencing (LTF knockdown, LTF-KD) A498 cells pretreated without or with recLTF protein at 300 ng/mL for 24 h. The different letters in each column of F and H indicate statistical significance at *p* < 0.05 as determined by the Friedman test.

**Figure 6 cancers-12-00847-f006:**
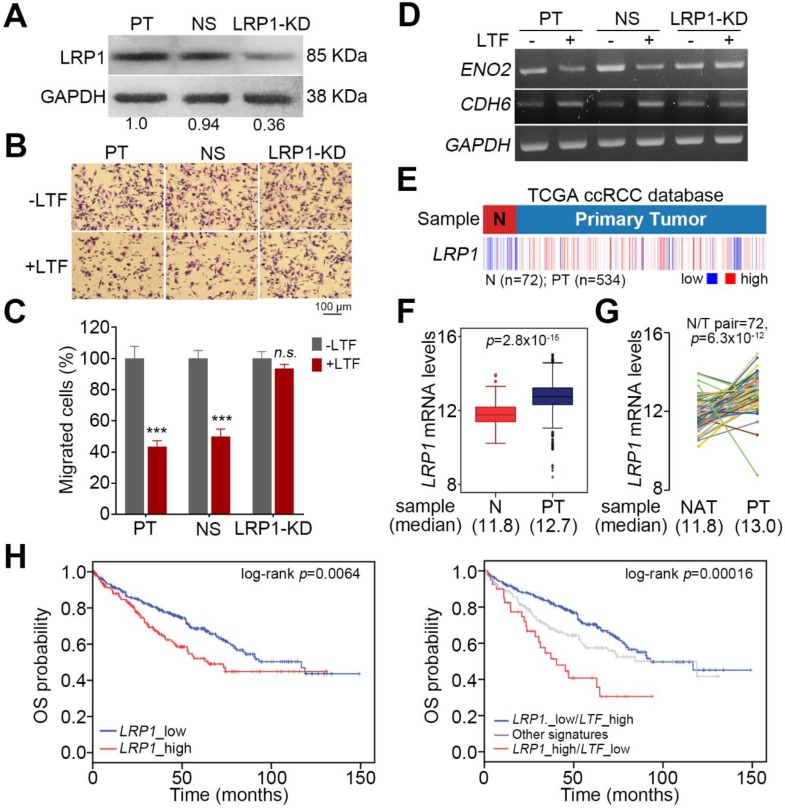
The targeting of LRP1 by extracellular LTF effectively suppresses the metastatic potential of ccRCC cells. (**A**) Western blotting analysis for LRP1 and GAPDH protein in ACHN cells without (PT) or with stable transfection of NS shRNA or LRP1 shRNA (LRP1-KD). (**B**,**C**) Giemsa staining (**B**) and histogram representing data from three independent experiments. (**C**) analyzing migrated cells after a 3-h Transwell assay in parental and vector control ACHN cells and LRP1-KD ACHN cells pretreated without or with recLTF protein at 300 ng/mL for 24 h. The nonparametric Mann–Whitney U test was used to estimate the statistical significance. (**D**) RT-PCR analysis for ENO2, CDH6 and GAPDH gene expression in parental and vector control and LRP1-KD ACHN cells treated without or with recLTF protein at 300 ng/mL for 24 h. (**E**,**F**) Heatmap (**E**) and boxplot (**F**) represent the transcriptional profile of the LRP1 gene in normal tissues (N) and primary tumors (PT) derived from TCGA ccRCC patients. Statistical significance was analyzed by t-test. (**G**) Gene expression of LRP1 in paired normal adjacent tissues (NATs) and primary tumors (PTs) derived from ccRCC patients in the TCGA database. Statistical significance was analyzed by a paired t-test. (**H**) Kaplan–Meier analyses of OS at a maximal risk condition for the mRNA levels of *LRP1* without [left; low (*n* = 344), high (*n* = 176)] or with combining LTF gene expression [right; LRP1_low/LTF_high (*n* = 275), other signatures (*n* = 205), and LRP1_high/LTF_low (*n* = 40)] in TCGA ccRCC patients.

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
