# Peer review of "Lactotransferrin Downregulation Drives the Metastatic Progression in Clear Cell Renal Cell Carcinoma"

_cancers, 2020, doi:10.3390/cancers12040847_

Round 1
Reviewer 1 Report
This is an interesting work dealing with the gene encoding lactotransferrin regulation disorders in patients with clear cell renal cell carcinoma (ccRCC). The clinical data and the molecular data were collected from the UCSC Xena website (http://xena.ucsc.edu/welcome-to-ucsc-xena/). The authors examined the transcriptional profile of LTF in normal tissues and primary tumors derived from The Cancer Genome Atlas (TCGA) patients with clear cell, chromophobe and papillary RCC. The data showed that LTF mRNA levels in primary tumors were significantly lower than those of normal tissues in the TCGA ccRCC dataset but not in TCGA chromophobe and papillary RCC datasets. The authors found that patients with ccRCC expressing a low level of LTF transcript had the shortest overall survival time. According to authors the results suggest that LTF may predict the outcome of ccRCC. Their findings may uncover a new mechanism by which LTF may interact with lipoprotein receptor-related protein 1(LRP1) to inhibit metastatic progression in ccRCC.
The work is well designed, concise and comprehensible. Charts and tables illustrate the results well. The only objection can be the concealment of clinical data of analyzed patients, which may be due to the fact that the data obtained from TCGA were analysed.
Author Response
Attached please find the responses to the comments and suggestions.

Reviewer 2 Report
- Please provide catalog number of antibodies used in this study.
- The human renal adenocarcinoma cell lines ACHN and A498 were used in this study. This manuscript indicated that the different LTF levels were found in the two cell lines. Authors have to introduce the different characteristics between ACHN and A498 and discuss the possible reasons about different experiment results found in both cell lines.
- Previous studies have demonstrated LTF was related to migration, apoptosis, cell cycle arrest in various cells. In this study, authors only indicated LTF was related to migration in ccRCC. Whether LTF did not affect apoptotic signals or cell cycle arrest signals in ccRCC? Authors have to explain or discuss at somewhere in this manuscript and show some experimental result about apoptosis or cell cycle. In addition, except A498, whether LTF can affect migration in other ccRCCs.
4. In figure 4, only mRNA levels were determined (E-G), whether authors can provide the data of protein levels.
Author Response

(The authors gave the same response as above.)

Reviewer 3 Report
- Introduction:
- sunitinib is no longer the standard of care for first line treatment. Immunotherapy (alone or in combination with TKIs) and cabozantinib. Please update your first paragraph.
- lines 77-82: please do not anticipate the results and conclusions. You have just to declare aim/objectives of your study.
- Results:
- which is the definition of low and high LTF? For each evaluation of LTF by IHC and by transcript level, please define how did you dichotomized the results and eventually median levels.
- Discussion:
- LTF maybe associated with inflammation, could different levels of LTF from tumor to normal tissue related to a different inflammatory state?
- Again, if LTF is associated with cancer-related inflammation, could also be a factor to be studied for prediction of response to immunotherapy (Brighi N. The Interplay between Inflammation, Anti-Angiogenic Agents, and Immune Checkpoint Inhibitors: Perspectives for Renal Cell Cancer Treatment. Cancers (Basel). 2019;11). Please discuss accordingly.
Author Response

(The authors gave the same response as above.)

Round 2
Reviewer 2 Report
The revised manuscript has replied the major questions.
Reviewer 3 Report
Authors answered to all the queries raised. I recommend paper acceptance in the present form.